# Acceptability and Practicality of a Quick Musculoskeletal Examination into Sports Medicine Pre-Participation Evaluation

**Stefano Palermi [1], Giada Annarumma [1], Alessandro Spinelli [1], Bruno Massa [1], Alessandro Serio [1], Marco Vecchiato [2], Andrea Demeco [3], Erica Brugin [4], Felice Sirico [1], Franco Giada [4] and Alessandro Biffi [5,*]**

[1] Public Health Department, University of Naples Federico II, 80131 Naples, Italy; stefanopalermi8@gmail.com (S.P.); giada.annarumma@gmail.com (G.A.); alessandro.spinelli1@gmail.com (A.S.); b.massa91@gmail.com (B.M.); alessandro.serio.1984@gmail.com (A.S.); sirico.felice@gmail.com (F.S.)

[2] Sports and Exercise Medicine Division, Department of Medicine, University of Padova, 35100 Padova, Italy; marcovecchiato.md@gmail.com

[3] Unit of Physical and Rehabilitative Medicine, Department of Medical and Surgical Sciences, University of Catanzaro Magna Graecia, 88100 Catanzaro, Italy; andreademeco@hotmail.it

[4] Cardiovascular Rehabilitation and Sports Medicine Service, Cardiovascular Department, Noale Hospital, 30033 Noale, Italy; erica.brugin@aulss3.veneto.it (E.B.); franco.giada@aulss3.veneto.it (F.G.)

[5] Med-Ex, Medicine & Exercise, Medical Partner Scuderia Ferrari, 00118 Rome, Italy

[*] Correspondence: alessandro.biffi@med-ex.it

**Abstract:** Background: Child musculoskeletal (MSK) diseases are common and, even if often benign, sometimes can lead to significant impairment in the future health of children. Italian pre-participation evaluation (PPE), performed by a sports medicine physician, allows for the screening of a wide range of children every year. Therefore, this study aims to evaluate the feasibility and the acceptability of pGALS (pediatric Gait, Arms, Legs and Spine) screening, a simple pediatric MSK screening examination, when performed as part of a routine PPE. Methods: Consecutive school-aged children attending a sports medicine screening program were assessed with the addition of pGALS to the routine clinical examination. Practicability (time taken) and patient acceptability (discomfort caused) were recorded. Results: 654 children (326 male, mean age 8.9 years) were evaluated through pGALS. The average time taken was 4.26 min (range 1.9–7.3 min). Acceptability of pGALS was deemed high: time taken was "adequate" (97% of parents) and caused little or no discomfort (94% of children). Abnormal MSK findings were common. Conclusions: pGALS is a practical and acceptable tool to perform in sports medicine PPE, even if performed by a non-expert in MSK medicine. Although common, abnormal MSK findings need to be interpreted in the global clinical context and assessment.

**Keywords:** musculoskeletal evaluation; pre-participation screening; children; sport medicine

## 1. Introduction

Musculoskeletal (MSK) diseases are far more common in children, even more than epilepsy and diabetes [1], and about one out of four children suffer from it [2]: MSK pain was identified as the second most reported physical symptom after headache in painful children [3]. Moreover, MSK complaints account for a big part of hospital charges and missed school days in the pediatric population [4]; they represent an important diagnostic challenge [5] since the diagnosis process in children is often difficult. Furthermore, there is a paucity of MSK research focusing on pediatric populations [6]. Usually, MSK symptoms in young subjects are benign, self-limiting, and trauma-related: specialist consultancies are not always necessary, and in many instances, reassurance alone may be sufficient. However, sometimes this symptomatology could be related to more serious diseases [7] (i.e., idiopathic juvenile arthritis, inflammatory bowel disease, cystic fibrosis,

Marfan syndrome, Ehler Danlos syndrome), or could have a significant impact on the future MSK health of children [8]: an early identification may offer a better chance of developing effective prevention and treatment strategies [9], in order to improve their prognosis.

Since 1982, it has been mandatory for every Italian competitive athlete to undergo an annual pre-participation evaluation (PPE) by sports medicine physicians, in order to obtain medical clearance for sport activities. PPE includes the assessment of the cardiovascular (CV) system and other non-CV investigations [10], and its main aim is to identify CV diseases that pose a risk of sudden death in the athlete [11]. This visit is free for underage athletes, according to Italian National Health System (NHS) rules. It implies that lots of apparently healthy young subjects are screened every year, with a huge public health role: for example, Vessella et al. [10] evaluated 5910 athletes (mean age 15.4 years) only in a year and in a single sports medicine center. Nevertheless, even if PPE has a great preventive role in cardiovascular screening, it has only a minor role in MSK assessment, except for some personal history questions. Indeed, even if MSK screening has a recognized important preventive function for the young athlete's health [12,13], currently guidelines mainly focus on the CV screening protocol for athletes [10]. Probably, the lack of MKS habits of Italian sports physicians, associated with the short time of a visit, are contributing factors to reducing the MSK role in the sports medicine PPE. However, this highlights a worrying scenario, since MSK injuries, even if less deadly compared to CV ones, are far the most frequent and preventable conditions affecting young athletes [14], given the risk of MSK injury in sports [15,16].

In recent years, several MSK screenings have been proposed in the literature, such as pediatric Regional Examination of the Musculoskeletal System (pREMS) [17], Functional Movement System (FMS) [18], or Gait, Arm, Leg Spine (GALS) [19]. GALS screening has gained particular scientific interest since it represents an easy and valid approach to adult MSK evaluation [20]. It has been demonstrated to have excellent sensitivity to detect MSK abnormality, incorporating simple maneuvers often used in clinical practice, and it is quick to perform, taking a few minutes to complete [21]. A pediatric version, pGALS, was validated in cohorts of children, especially in pediatric, rheumatology and emergency settings: it was able to rapidly diagnose MSK abnormalities [21], even if performed by non-MSK specialists [20]. pGALS is the modified version of adult GALS, adapted for school-age children. It is very similar to adult GALS in format and content but differs in the simplicity of the commands to be executed and in some more specific items for the foot, ankle, and TMJ (temporomandibular joint) [22].

Considering that pGALS has been validated worldwide [23–26] and has shown great sensibility and specificity as an MSK diagnostic tool [27], the aim of the present paper was to evaluate the acceptability and practicability of pGALS in the Italian sports medicine PPE.

## 2. Materials and Methods

The setting of this study was the annual "Ferrari Formula Benessere" program, a corporate wellness program conducted by the Med-Ex Medicine and Exercise society. The program includes several medical activities directed at adults, women and children and it is firmly focused on the concept of primary prevention and exercise prescription in the general population. Its effectiveness has been already proved [28–32]. During the program, a specific 4-week period is dedicated to the medical evaluation of employers' children. Each child underwent several medical evaluations carried out by medical specialists in different disciplines (i.e., sports medicine physicians, cardiologists, pediatrics, orthopedics, dermatologists, nutritionists and others). Participation in the program is voluntary and totally free both for employers and their relatives.

Therefore, the inclusion criteria for the study were children from 5 to 16 years, attending the sports medicine visit of the Med-Ex screening. The exclusion criteria were any history of musculoskeletal, neurological, or orthopedic disorders of the child. Over a

four-week period between November and December 2021, all eligible subjects were invited to take part in the study. Children were assessed as per routine clinical visit by the sports medicine doctor on duty and if eligible were approached to take part in the study, along with their parents.

The sports medicine visit was organized in three parts: personal and family anamnesis collection, physical examination and resting electrocardiogram. After appropriate consent was obtained, pGALS screening was performed by the sports medicine physician at the beginning of the physical examination part of the visit. pGALS is a simple and quick musculoskeletal assessment to distinguish between abnormal from normal findings in children and young people [20]. It is organized into 19 sections that assess pain, limitation of movement and joint disease in posture, upper limbs, lower limbs and the spine. It consists of a series of simple maneuvers to assess all major joints to discern normal from abnormal findings (Table 1) and it does not require any materials to be performed: based on the presence or absence of MSK disorders, all the sections of the questionnaire are fulfilled.

**Table 1.** The pGALS assessment.

| Category | Screening Maneuvers | What Is Being Assessed? |
|---|---|---|
| Gait | ➢ Observe the child walking<br>➢ Observe the child walking on their heels<br>➢ Observe the child walking on their tiptoes | ➢ Ankle, subtalar, midtarsal and small joints of feet and toes<br>➢ Foot posture |
| Arm | ➢ "Put your hands out in front of you"<br>➢ "Turn your hand over and make a fist"<br>➢ "Pinch your index finger and thumb together"<br>➢ "Touch the tips of your fingers with your thumb"<br>➢ "Squeeze metacarpophalangeal joints"<br>➢ "Put your hands together palm to palm"<br>➢ "Put your hand together back-to-back"<br>➢ "Reach up and touch the sky"<br>➢ "Look up at the ceiling"<br>➢ "Put your hand behind your neck" | ➢ Neck extension<br>➢ Forward flexion of shoulders<br>➢ External rotation of shoulders<br>➢ Shoulder abduction<br>➢ Elbow extension/flexion/supination<br>➢ Wrist extension/supination<br>➢ Metacarpophalangeal joints<br>➢ Extension/flexion of small joints of fingers<br>➢ Manual dexterity<br>➢ Coordination of small joints of index finger and thumb and functional key grip |
| Leg | ➢ Patella tap and cross fluctuation<br>➢ "Ben and then straighten your knee"<br>➢ Passive flexion with internal rotation of the hip | ➢ Knee effusion<br>➢ Knee flexion/extension<br>➢ Hip flexion and external rotation |
| Spine | ➢ "Open your mouth and put 3 of your fingers in your mouth"<br>➢ "Try and touch your shoulder with your ear"<br>➢ Observe spine from behind | ➢ Posture and habitus<br>➢ Temporomandibular joints<br>➢ Cervical spine lateral flexion<br>➢ Forward flexion of thoraco-lumbar spine |

> ➢ "Can you bend and touch your ➢ Scoliosis
> toes?"

A dedicated online form was used for the study to collect data including patient demographics and details of pGALS assessment. The time necessary to perform pGALS maneuvers was recorded.

The methods used are similar to those used previously to assess the acceptability and practicality of pGALS [20,23,33]. Parents were invited to complete a 0–5 Likert scale assessment of the acceptability of pGALS in terms of the time taken for the examination, with a scale of 5 options (ranging from "far too short" to "far too long"), while children were asked similarly about the discomfort of the visit itself, with a scale of 5 options (ranging from "lots of pain" to "no pain") [20].

Before the visit, each patient signed an informed consent accepting medical procedures and data collection by Med-Ex. All information was recorded anonymously. Moreover, the data collection form specifies that data should be used for scientific purposes, in aggregate form and maintaining the privacy of each specific subject. Med-Ex treats these data according to privacy rules and protection. All procedures performed in studies were in accordance with the Helsinki declaration and its later amendments or comparable ethical standards

*Statistical Analysis*

All analyses were performed by an author not involved in data collection and were conducted using Stata Software v.12 (StataCorp. 2011, Stata Statistical Software: Release 12, College Station, TX, USA: StataCorp LP). Statistical analysis used nonparametric tests with primary outcomes being the practicality (completeness) and acceptability (time taken and degree of discomfort) of the pGALS assessment. The value of $p < 0.05$ was considered statistically significant.

### 3. Results

The study group included 654 children (326 male, 328 female). The mean age was 8.9 ± 2.8 years. The pGALS examination was completed in all children (654, 100%).

The average time taken for the pGALS examination was 4.26 min (range 1.9–7.3 min). All study participants completed the acceptability questionnaire, with most parents (637/654, 97%) stating that the time taken was ''adequate'' (Figure 1). Data on perceived discomfort reported that most children (621/654, 94%) felt no pain during the visit (Figure 2): the only pain reported was that related to the eventually painful joints.

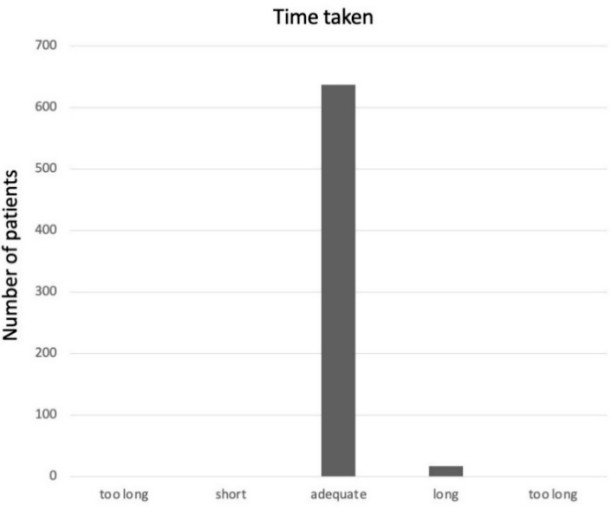

**Figure 1.** Acceptability of pGALS about the time taken (on the Likert scale).

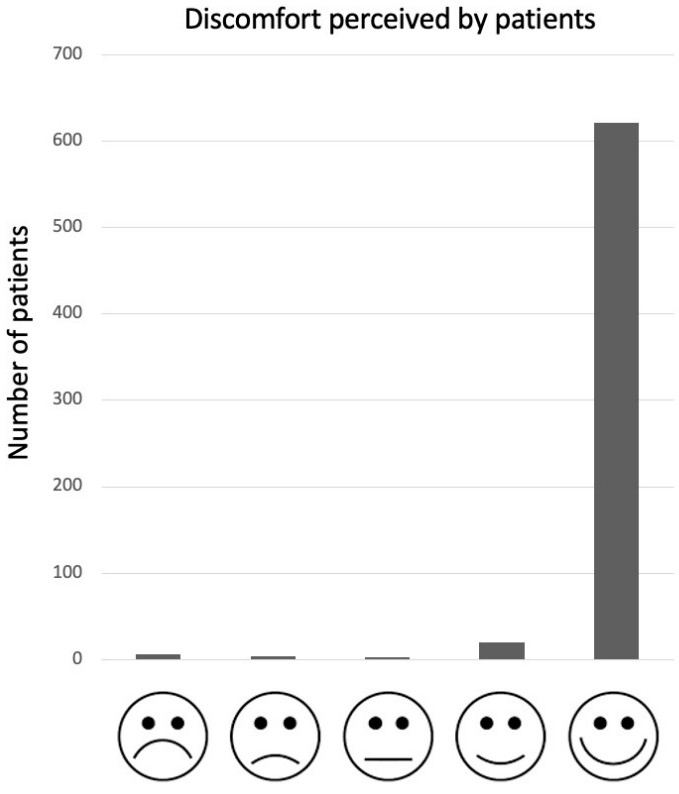

**Figure 2.** Acceptability of pGALS about discomfort perceived (on the Likert scale).

Table 2 shows the details of abnormal findings of pGALS screening. About forty percent of subjects tested abnormal at pGALS screening, with gait as the most common abnormal part. Details are reported in the last column.

**Table 2.** Abnormal findings of pGALS screening.

| Component of pGALS Screen | Number of Abnormal Subjects | Most Frequent Abnormal Detail |
|---|---|---|
| Gait | 152 (23.2%) | ➢ Flat feet<br>➢ Heel pain |
| Arms | 15 (2.3%) | ➢ Hyperlaxity ligaments |
| Legs | 33 (5.0%) | ➢ Tibial plateau pain<br>➢ Genu valgum/varum |
| Spine | 82 (12.5%) | ➢ Suspected scoliosis<br>➢ Dorsal kyphosis |

### 4. Discussion

Results of the present study show that pGALS screening can be effectively performed in PPE by sports medicine physicians and that it is acceptable to both parents and children. This important finding supports the use of MSK screening as a core clinical skill in routine sports medicine clinical evaluation.

Since MSK problems are very common during childhood, their screening methods should be sensitive, practical, easy to apply, and acceptable for both children and families. We demonstrated that a complete pGALS screen can be easily performed on all children as part of the first part of the sports medicine visit, in the physical examination stage, with a short time and without particular instruments.

The two critical items analyzed are time spent and the discomfort perceived by children.

The pGALS examination was completed in about 4 min. This data is comparable to the time taken in previous reports by non-experts in MSK disease [34] but longer than the time taken by pediatric rheumatologists or primary care doctors [33,34]. Obviously, the pGALS examination takes slightly longer (about 6 min) if abnormal findings are observed. We believe that adequate training of the physician both in performing maneuvers and in recognizing pathological findings could further shorten the time taken for the pGALS performance, making it easily integrated into routine clinical practice. Indeed, interpretation of MSK examination in children requires knowledge of normal growth variants to facilitate appropriate interpretation, and this is an important aspect to consider.

Researchers performing pGALS in our study were sports medicine physicians, and their training for this study was based mostly on pGALS video demonstrations. Whilst using pGALS, the verbal instructions were supplemented by a 'copy me' approach by the physician, as performed in previous research [23], and this may have facilitated understanding since our sample size was composed mainly of young children, and many instructions could have been misinterpreted by them ("Squeeze metacarpophalangeal joints"). Indeed, medical communication with children is a difficult skill for a non-pediatric specialist [35], and this is an important aspect to consider. Moreover, pediatric medical evaluation has some difficulties: as suggested by Foster et al. [27], in this population it is fundamental to look for verbal and non-verbal signs of discomfort, which may hint at further pathologies (such as facial expression, withdrawal of limb, or refusal to be examined further). It is therefore important that all clinicians to whom children with MSK problems may present have the necessary skills to effectively triage patients, and where appropriate instigate referral to specialist services. Moreover, many doctors, including Italian sports medicine physicians report a lack of confidence and competence in their MSK [36,37] clinical skills, often due to a lack of this teaching at undergraduate and postgraduate levels. We suggest that these key skills and knowledge should be included in their training courses, including sports medicine residency teaching.

Knowledge of the normal range of motion of joints in different ethnicities and ages is essential to properly interpret pGALS, and the examiner must carefully look for asymmetries and slight changes. Furthermore, observers must know normal variants in gait, leg alignment and motor milestones [27], as these often worry parents, especially regarding pre-school children, that therefore have to be educated and reassured. Normal variants are not painful and so it is important not to associate any MSK symptoms with normal variants.

The Likert scale shows that almost all of the parents considered the time spent to perform pGALS as appropriate. Therefore, the reaction of parents to the sports medicine visit conducted with the addition of the pGALS maneuvers was positive, as it was perceived as further attention to their children: they wanted their child to be examined as thoroughly as possible.

Moreover, in our study, almost all of the children report few discomforts on the Likert scale, confirming that the pGALS examination is acceptable also to the athlete. This disagrees with similar studies performed in acute settings, since some of them, notably those presenting with joint or abdominal pain, complained of significant discomfort during the pGALS examination [20,34]. Our sample size was composed of apparently healthy children requiring sports medicine clearance to engage in sport activities, so it is likely that our children had no pain at the moment of the visit.

This is the first study to validate the use of pGALS in an Italian population of predominantly healthy children. We acknowledge that our findings on pGALS were not validated by MSK 'experts' (e.g., physical medicine and rehabilitation doctors or orthopedic surgeons) but we regard our findings as useful to assess practicality and acceptability rather than the validity of pGALS itself. Indeed, the sensibility and specificity of this screening maneuver have already been proved to be high, with values ranging around 90% [25,26]: Foster and Jandial [27] reported a sensitivity of 97% and a

specificity of 98% for pGALS, Moreno-Torres et al. [25] showed that pGALS had a sensitivity of 97% and specificity of 93% in Mexican Spanish translation, Batu et al. [26] had the sensitivity slightly lower (93.7%) and a specificity of 97.4% in Turkish children, and finally, Sukharomana et al. [24] found a sensitivity and specificity of 74.14% and 100% in a Thai population.

Although not a primary objective of the study, we have demonstrated that within the recruited sample of apparently healthy children, MSK presentations and abnormalities in the performance of pGALS screening are common (about 40% of all screened subjects). However, the abnormalities in pGALS did not always mean an underlying MSK pathology. Clearly, the overlap between MSK and other systems is important and may lead to ''false-positive'' results in the pGALS screening examination. Indeed, pGALS was developed in the context of detecting inflammatory joint disease but has been shown to be useful in identifying other joint problems (e.g., orthopaedic problems at the hip, scoliosis, hypermobility), joint involvement in mucopolysaccharidoses [38], as well as other pathologies (stroke and sepsis) [33,34]. This emphasizes the need for pGALS to be a part of the global clinical assessment and the findings to be interpreted in the context of the clinical scenario, especially in a sports medicine PPE.

Flat feet and suspected scoliosis were some of the more frequent detected conditions, and these are commonly found conditions in most PPE [39,40]: screening of these features is crucial in order to avoid further problems in children's future health, other than possible impairments in the carrier of an athlete. Therefore, we want to suggest a list of not-to-miss MSK conditions in every sports medicine PPE, that could have a worse prognosis for athlete carriers if not promptly diagnosed (Table 3) and that could benefice specific treatments. Following pathological findings found on pGALS screening, the observer should be directed to a more detailed examination of the relevant area, often with the help of an MSK-specialist physician.

**Table 3.** Important musculoskeletal (MSK) conditions to search in the sports medicine pre-participation evaluation (PPE).

| Category | What Can Be Pathological | Outcome on Future Health, If Not Treated |
|---|---|---|
| Gait | Flat feet [40] | ➢ Feet and ankle pain<br>➢ Ankle instability |
| | Pes cavus | ➢ Forefoot pain<br>➢ Lower limb tendinopathy<br>➢ Plantar fasciitis<br>➢ Ankle instability |
| | Heel Pain (suspected Sever Blank disease) [41] | ➢ Longer recovery times<br>➢ Impaired sport activities<br>➢ Calcaneal apophysis fracture |
| Arm | Hyperlaxity of ligaments [42] | ➢ May need surgery<br>➢ Neurodevelopment impairment<br>➢ Under-diagnosed related disorder<br>➢ Joint dislocations and sprains |
| Leg | Genu valgum/genu varum [43] | ➢ Early osteoarthritis of knee joint<br>➢ Meniscus injury<br>➢ Ligament injury<br>➢ Iliotibial tendinopathy<br>➢ Gait impairment<br>➢ Knee instability<br>➢ Knee flexion/extension<br>➢ Hip flexion and external rotation |
| | Tibial plateau pain (suspected Osgood Schlatter disease) [44] | ➢ Long-term pain |

| Spine | Scoliosis [45] | | ➤ Impairment on respiratory function |
|---|---|---|---|
| | | ➤ Bone deformation | |
| | | ➤ Growth plate abnormalities | |
| | | ➤ Patellar tendon pathology | |
| | | ➤ Back pain | |
| | | ➤ Arthrosis | |
| | | ➤ Back muscle weakness | |
| | | ➤ Digestive problems | |
| | | ➤ May need surgery | |
| | Dorsal kyphosis (suspected Scheuermann disease) [46] | ➤ Back pain | |
| | | ➤ Neck pain | |
| | | ➤ May need surgery | |

This study suffers from some limitations. First, our protocol did not include verification of the sports medicine's findings by an MSK expert physician, such as an orthopedic or a physical medicine and rehabilitation doctor. Moreover, interpretation of the pGALS examination requires knowledge of normal variants and assessment of the child in discomfort to facilitate appropriate interpretation: these skills may be difficult to include in routine physician's clinical teaching. Finally, different from similar studies [24,26], we did not use an Italian translation (and thus, validation) of the PGALS questionnaire.

Future studies should aim to compare the results obtained through different learning methods for the performers of pGALS in clinical and preventive contexts in order to ensure that any healthcare figure, especially sports medicine physician, could be able to effectively perform an efficient MSK screening evaluation. Moreover, in the MSK routine screening, there could be other potential skills to evaluate, such as balance, proprioception and strength: in that way, tests such as single-leg stance and squat jump could be very useful to incorporate into the screening maneuvers [47,48], in order to have a multidisciplinary quick evaluation of the athlete.

**5. Conclusions**

Results of the present study provide evidence that pGALS is practical, acceptable, and useful as a clinical skill to be incorporated, and clinically interpreted, into regular sports medicine PPE with the ultimate aim to facilitate earlier diagnosis of children with MSK diseases and optimize their outcome.

**Author Contributions:** S.P., M.V. and B.M. conceived of the presented idea. G.A., A.S. (Alessandro Spinelliand) and E.B. developed the theory and performed the computations. F.S., A.B., A.S. (Alessandro Serio), F.G. and A.D. verified the analytical methods. All authors discussed the results and contributed to the final manuscript. All authors have read and agreed to the published version of the manuscript.

**Funding:** This research received no external funding.

**Institutional Review Board Statement:** The study was conducted in accordance with the Declaration of Helsinki.

**Informed Consent Statement:** Informed consent was obtained from all subjects involved in the study.

**Data Availability Statement:** Not applicable.

**Conflicts of Interest:** The authors declare no conflict of interest.

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
