# Peer review of "Acceptability and Practicality of a Quick Musculoskeletal Examination into Sports Medicine Pre-Participation Evaluation"

_pediatrrep, doi:10.3390/pediatric14020028_

Round 1

Reviewer 1 Report

The manuscript addresses an interesting topic of pre-participation screening evaluation for kids. It is meaningful tool to detect their musculoskeletal problems in earlier stage. The aim of the study was to evaluate the acceptability and practicability of pGALS. The study has several strengths, but there are a few topics that could be better explored. I made a few comments and suggestions for each section of the manuscript, which is shown below.

Page

Line

Comments and Recommendations

2

59

Ø  Probably, the lack of MKS habits 59

MSK?

If you are OK, why don’t you abbreviate musculoskeletal as MS just like cardiovascular (CV)?  For me, it will be easier to read.

3

Table 1

Ø  waling

Walking?

Please, double check mistakes in spelling and grammar throughout the manuscript.

If you can, make which screening maneuvers belong to which category.

Does your participants understand “Squeeze metacarpophalangeal joints” without help?

Can you mention what is the difference between the pGALS and the adult-version GALS?

4

134

Ø  Data on per-134 ceived discomfort reported that most children (621/654, 94%) felt no pain for the visit (Fig-135 ure 2). 136

Can you add where was the pain and discomfort reported during which item?

6

Table 2

Ø  Total 282 (43,1%)

Were all abnormal findings from different participants? If not, don’t just add up all.

The current version of the manuscript has only descriptive results, more in-depth analysis of the data could strengthen the application of study.

6

150

Ø  Since MSK problems are very common during childhood, their screening methods 150 should be sensitive, practical, easy to apply, and acceptable for both children and families. 151

Can you compare the result to the adult-version PALS?

Is there any screening item to add or delete to be more effective and efficient? What do you think add something more challenging, like jumping?

Author Response

Dear reviewer,

please find attached our answers to your valuable suggestions

Reviewer 2 Report

Dear authors, you can find the considerations and suggestions for improving your work below.
Best regards.

In the Introduction: I find it interesting to add some data on the number of young athletes who must pass this medical examination each year in Italy.

In the Materials and Methods:

  • Were there any exclusion criteria?
  • Is some type of material necessary to correctly apply pGALS? For example, a stretcher...
  • Has the study been accepted by an ethics committee?

In the discussion, line 159: What was the mean time of the visits in which abnormal findings were found?

Finally, I think it would be good to add a section with the limitations of the study.

Author Response

(The authors gave the same response as above.)
